# The *Paeonia qiui* R2R3-MYB Transcription Factor PqMYBF1 Positively Regulates Flavonol Accumulation

**DOI:** 10.3390/plants12071427

**Published:** 2023-03-23

**Authors:** Yue Zhang, Jingjing Duan, Qiaoyun Wang, Min Zhang, Hui Zhi, Zhangzhen Bai, Yanlong Zhang, Jianrang Luo

**Affiliations:** 1College of Landscape Architecture and Arts, Northwest A&F University, Yangling 712100, China; 2National Engineering Research Center for Oil Peony, Yangling 712100, China

**Keywords:** *Paeonia qiui*, leaf color, flavonol, transcription factor, MYB

## Abstract

Tree peony is a “spring colored-leaf” plant which has red leaves in early spring, and the red color of the leaves usually fades in late spring. Flavonols are one subgroup of flavonoids, and they affect the plant organs’ color as co-pigments of anthocyanins. To investigate the color variation mechanism of leaves in tree peony, *PqMYBF1*, one flavonol biosynthesis-related MYB gene was isolated from *Paeonia qiui* and characterized. PqMYBF1 contained the SG7 and SG7-2 motifs which are unique in flavonol-specific MYB regulators. Subcellular localization and transactivation assay showed that PqMYBF1 localized to the nucleus and acted as a transcriptional activator. The ectopic expression of *PqMYBF1* in transgenic tobacco caused an observable increase in flavonol level and the anthocyanin accumulation was decreased significantly, resulting in pale pink flowers. Dual-luciferase reporter assays showed that PqMYBF1 could activate the promoters of *PqCHS*, *PqF3H,* and *PqFLS*. These results suggested that PqMYBF1 could promote flavonol biosynthesis by activating *PqCHS, PqF3H,* and *PqFLS* expression, which leads metabolic flux from anthocyanin to flavonol pathway, resulting in more flavonol accumulation. These findings provide a new train of thought for the molecular mechanism of leaf color variation in tree peony in spring, which will be helpful for the molecular breeding of tree peony with colored foliage.

## 1. Introduction

Leaf color is one of the attractive traits for ornamental plants, and plants with colored foliage are often called “colored-leaf plants”, including spring colored-leaf plants, autumn colored-leaf plants, and colored-leaf plants all year round. Among them, spring colored-leaf plants can be used in landscape construction in early spring. The red leaf color formation in early spring and the red color fading of leaf in late spring are two remarkable events in spring colored-leaf plants, such as tree peony and crabapple.

Studies have shown that the color variation from red to green is caused by the change of pigments inside the leaves. It is well known that red color usually arises from anthocyanins (one subgroup of flavonoids). For example, Tang et al. reported that the anthocyanin content gradually decreases during red fading in herbaceous peony leaves [1]. Decreasing anthocyanin content led to fading red color in *Malus* ‘Radiant’ leaves [2]. Flavonols, another subgroup of flavonoids, can also modulate plant organ color as co-pigments of anthocyanins [3,4].

As a branch of flavonoid biosynthesis, the biosynthesis of flavonol has been studied in *Arabidopsis*, *Vitis vinifera*, *Malus crabapple,* and other species [5,6,7,8,9,10]. First, chalcone synthase (CHS) catalyzes coumaroyl-CoA and malonyl-CoA to form chalcone, which is isomerized to naringenin by chalcone isomerase (CHI). Then, naringenin is converted to dihydroflavonols by flavanone-3-hydrolase (F3H) and flavonoid 3′-hydroxylase (F3′H). Subsequently, dihydroflavonols can be converted to anthocyanins by dihydroflavonol 4-reductase (DFR) and anthocyanidin synthase (ANS). At the same time, flavonols can also be formed from didydroflavonols by flavonol synthase (FLS). The competition relationship between FLS and DFR directly affects the accumulation of flavonols and anthocyanins, which in turn affects plant coloration [4,11,12,13].

In addition, the biosynthesis of flavonol is also controlled by a complex regulatory network, the most important of which is the MYB regulator [14,15,16]. In *A. thaliana*, R2R3-MYBs are divided into 25 subfamilies according to the differences in C-terminal. MYBs in subgroup SG4, SG5, SG6, SG7, and SG15 are involved in regulating the synthesis of flavonoids [17,18]. MYB members belonging to the SG7 group play a significant role in the synthesis of flavonol [19]. In recent years, R2R3-MYB transcriptional activators for flavonol synthesis have been isolated from *Gerbera hybrida*, *Medicago truncatula*, *Malus sieversii*, *Fagopyrum tataricum,* and *Freesia hybrida* [20,21,22,23,24]. For example, the pear PbMYB12b protein positively regulates the accumulation of flavonol by promoting the expression of *PbCHSb* and *PbFLS* [25]; the overexpression of *Mt134* from *M. truncatula* in *Arabidopsis* complements the flavonol deficiency [23]. In addition to transcriptional activators, researchers have also isolated repressors that inhibited flavonol biosynthesis, such as FaMYB1, from strawberry (*Fragaria × ananassa*) [26]. In *Chrysanthemum × morifolium*, CmMYB012 inhibits flavonol biosynthesis by inactivating the transcriptional activity of *CmCHS* [27]. However, most of these studies were conducted in flowers or fruits; the regulation mechanism of flavonol biosynthesis in spring colored-leaf plants is not clear.

Tree peony is one of the top ten Chinese traditional flowers and is known as the “king of flowers” for its gorgeous and magnificent flowers. At the same time, tree peony is also a “spring red leaf” plant. Its young leaves are purplish red in early spring, and the red color of the leaves usually fades in late spring [28]. Among all kinds of tree peony, the leaf coloring trait of *Paeonia qiui* is the most typical. In *P. qiui*, the high level of anthocyanins rapidly accumulates in red young leaves, after which anthocyanin level decreases and other flavonoids (including flavonol) increase with leaf development [29]. In our previous studies, some flavonoid biosynthesis-related R2R3-MYBs were found in *P. qiui* through transcriptome sequencing [30]. Among them, two R2R3-MYBs (PqMYB113 and PqMYB4) were isolated, and their biological functions were verified in anthocyanin biosynthesis [31,32]. In this study, another flavonoid (flavonol) biosynthesis activator, PqMYBF1 from the leaves of *P. qiui*, was characterized, and the biological function of PqMYBF1 was investigated using genetically modified tobacco, which will enrich our knowledge about tree peony leaf color variation in spring.

## 2. Results

### 2.1. PqMYBF1 Is a Potential Regulator of Flavonol Biosynthesis

Phylogenetic analysis showed that PqMYBF1 was placed into the subgroup 7 (SG7) family proteins which are flavonol-specific R2R3-MYB transcription factors (TFs), such as grape VvMYBF1, apple MdMYB22, tomato SlMYB12, Arabidopsis AtMYB11, AtMYB12, and AtMYB111. PqMYBF1 shared 44.29% amino acid sequence identity with grape VvMYBF1, 35.95% with Arabidopsis AtMYB12 and 35.48% with apple MdMYB22 (Figure 1A). Based on these results, we speculated that PqMYBF1 was probably involved in the regulation of flavonol synthesis of leaves in *P. qiui*.

The ORF of *PqMYBF1* was 1140 bp encoding a protein of 379 amino acid residues. Multiple sequence alignment analysis revealed that PqMYBF1 contains the typical SANT domain of the MYB family at the N-terminus (Figure 1B). The SG7 motif ([K/R][R/x][R/K]xGRT[S/x][R/G]xx[M/x]K) and SG7-2 motif ([W/x][L/x]LS) which are the characteristics of flavonol biosynthesis regulators, were also found at the C-terminus of PqMYBF1 (Figure 1B) [7,8,19,33,34]. PqMYBF1 did not contain the motif [D/E]Lx2[R/K]x3Lx6Lx3R that interacts with bHLH proteins, suggesting that PqMYBF1 functions independently without bHLH [35].

### 2.2. PqMYBF1 Localizes to the Nucleus and Acts as a Transcriptional Activator

To investigate the subcellular localization of PqMYBF1, the recombinant vector pCAMBIA2300-*PqMYBF1*-GFP was transformed into onion epidermis. Red fluorescence showed the location of the nucleus. Onion cell expressing the PqMYBF1-GFP fusion protein showed a strong fluorescent signal in the nucleus. Therefore, we speculated that PqMYBF1 was localized and functioned in the nucleus (Figure 2A).

A transcriptional activity test was performed in Y2H yeast using a recombinant vector pGBKT7-*PqMYBF1* carrying the GAL DNA-domain to determine whether PqMYBF1 has transcriptional activation activity. Yeast carrying pGBKT7-*PqMYBF1* grew normally and appeared blue on SD/-Trp with X-α-gal adding aureobasidin A (AbA). The negative control group showed an inhibition of colony growth (Figure 2B). These indicated that PqMYBF1 has transcriptional activation activity.

### 2.3. PqMYBF1 Expression Correlates with Flavonol Accumulation and Flavonol Biosynthetic Gene Expression

The expression levels of *PqMYBF1* and flavonoid biosynthetic genes in different leaf stages were revealed by qRT-PCR. The content of anthocyanins and flavonols were measured (Figure 3). The total anthocyanin content of *P. qiui* leaves increased first, peaked at S3, and then decreased mildly in S4 and declined dramatically in S5, which was consistent with the observed phenotype (Figure 3B). For flavonol content, the results showed that there was a slight increased trend from S1 to S3, and then reduced slightly at S4 and S5, which overlapped with that of anthocyanin, but was not the same. The flavonol content was the highest in S3 stage, followed by S4, S5, and S2; the lowest was in S1 (Figure 3C).

As for the expression levels of *PqMYBF1* (Figure 3D), the results showed that it increased first and then decreased. *PqMYBF1* had the highest expression level in the S3 stage. In addition, the expression levels of flavonoid biosynthetic genes were also analyzed. The result showed that *PqCHS*, *PqCHI*, *PqF3H*, *PqF3’H*, *PqFLS*, *PqDFR*, and *PqANS* genes presented a basically consistent trend, which increased first and then decreased. The maximum value of their expression was in S3 stage. The expression levels of *PqCHI* were the lowest in S1, while the expression levels of *PqCHS*, *PqF3H*, *PqF3’H*, *PqFLS*, *PqDFR*, and *PqANS* were the lowest in the S5 (Figure 3D).

### 2.4. Overexpression of PqMYBF1 Promoted Flavonol Accumulation and the Expression of Flavonol Pathway Genes in Tobacco

To further investigate the function of *PqMYBF1*, the gene was cloned into an overexpression vector driven by 35S promoter and overexpressed in tobacco. Thirteen overexpression (OE) transgenic lines were generated (Figure 4A). Three transgenic lines (OE-6, OE-8 and OE-11) were selected for subsequent experiments. The flowers of three transgenic tobacco lines showed pale pink color, whereas the wild type (WT) lines showed red-pink flowers (Figure 4B). Then, the level of total flavonols and anthocyanins in transgenic and the WT petals were quantified. The total flavonol contents of the transgenic lines were significantly higher by 1.68- (OE-6), 1.57- (OE-8) and 3.17-fold (OE-11) than those in the WT. However, the anthocyanin contents of the transgenic plants were decreased by 71.16% (OE-6), 60.17% (OE-8) and 86.97% (OE-11) compared with the WT (Figure 4C). qRT-PCR was performed to evaluate the expression of anthocyanin and flavonol biosynthesis pathway genes in transgenic tobacco flowers. *PqMYBF1* was overexpressed in transgenic tobacco flowers and absent in the WT. Compared with the WT, the expression levels of *NtCHS*, *NtF3H*, *NtF3′H*, and *NtFLS* in transgenic tobacco flowers were higher, especially *NtCHS*, *NtF3H*, and *NtFLS* (Figure 4D). However, the expression levels of *NtDFR* and *NtANS* decreased slightly compared with the WT. 

We also investigated the influence of *PqMYBF1* on flavonol synthesis in tobacco leaves. As shown in Figure 5, the flavonol content of tobacco leaves in OE-6, OE-8, and OE-11 were about 3.4, 2.0, 3.6 times those in wild type leaves, respectively. We quantified the flavonol-related genes expression of transgenic and wild type tobacco leaves. *PqMYBF1* gene was highly expressed in transgenic tobacco lines leaves and had little or no expression in the WT. The expression levels of *NtCHS*, *NtCHI*, *NtF3H*, and *NtFLS* in transgenic tobacco leaves were much higher than those in the WT, especially *NtCHS* (up-regulated by 8.24-fold in OE-6, 5.83-fold in OE-8, and 27.5-fold in OE-11), *NtF3H* (up-regulated by 7.68-fold in OE-6, 4.69-fold in OE-8, and 51.71-fold in OE-11)*,* and *NtFLS* (up-regulated by14.67-fold in OE-6, 12.57-fold in OE-8, and 43.57-fold in OE-11) (Figure 5). All of these suggested that PqMYBF1 could promote flavonol biosynthesis and accumulation in transgenic tobacco. 

### 2.5. PqMYBF1 Activated the Promoters of PqCHS, PqF3H, and PqFLS

The expressions of *NtCHS*, *NtF3H*, and *NtFLS* genes were more strongly increased than other genes of the flavonol pathway in *PqMYBF1*-overexpressing tobacco lines. We speculated that *PqCHS*, *PqF3H*, and *PqFLS* were potential targets of PqMYBF1 transcriptional activation. A dual-luciferase reporter assay was carried out to determine the interaction of PqMYBF1 with *PqCHS*, *PqF3H*, and *PqFLS*. The promoter sequences of *PqCHS, PqF3H,* and *PqFLS* from *P. qiui* were cloned and named as *pPqCHS, pPqF3H,* and *pPqFLS*, and the length of them were 2016 bp, 1624 bp, and 1488 bp, respectively. 

The key *cis*-elements of *pPqCHS*, *pPqF3H*, and *pPqFLS* analysis were carried out using the PlantCARE online software and MYB binding sites were found in all of these three promoters (Figure 6A). These results indicated that *pPqCHS*, *pPqF3H*, and *pPqFLS* may be subject to MYB regulation. These three promoters were cloned into pBI121 vector with GUS reporter gene. GUS staining results showed that tobacco leaves treated with the fusion expression vectors and pBI121-*pPqCHS*-GUS, pBI121-*pPqF3H*-GUS, and pBI121-*pPqFLS*-GUS showed different degrees of staining, indicating that the promoters of *PqCHS*, *PqF3H*, and *PqFLS* genes have biological activity **(**Figure 6B).

Dual-luciferase reporter assay revealed that the promoter luminescence intensities of *PqCHS*, *PqF3H*, and *PqFLS* were all significantly increased by PqMYBF1 compared with the corresponding controls (Figure 7). This indicated that PqMYBF1 could activate the promoters of *PqCHS*, *PqF3H*, and *PqFLS*.

## 3. Discussion

It is well known that the TFs in the same subgroup usually have similar function [36]. In plants, R2R3-MYB TFs can be categorized into at least 25 subgroups, of which subgroup 7 plays a vital role in the synthesis of flavonol, which is characterized by the SG7 motif ([K/R][R/x][R/K]xGRT[S/x][R/G]xx[M/x]K) and the SG7-2 motif ([W/x][L/x]LS) [19]. Phylogenetic analysis indicated that PqMYBF1 was clustered within subgroup 7, with other known flavonol regulators such as AtMYB11, AtMYB12, and AtMYB111 from Arabidopsis, GtMYBP3, and GtMYBP4 from gentian and VvMYBF1 from grape (Figure 1A) [7,8,19,37]. Sequence analysis showed that PqMYF1 not only contains R2 and R3 domain, but also includes the SG7 motif and the SG7-2 motif (Figure 1B). In addition, no [DE]Lx(2)[RK]x(3)Lx(6)Lx(3)R motif for interaction with bHLH proteins was found in PqMYBF1, indicating that PqMYBF1 is functionally independent on bHLH cofactors. So far, no cofactors have been reported as being necessary for plant flavonol-specific MYBs [5]. All of these indicated that PqMYBF1 should be a potential R2R3-MYB protein that has the function of regulating flavonol biosynthesis.

Previous studies have shown that flavonol accumulation was promoted by flavonol biosynthesis MYB regulator [36,38]. In grape, the expression level of *VvMYBF1* was closely related to the accumulation of flavonol in grape berries. Overexpression of *MdMYB22* in apple callus significantly increased flavonol accumulation [20]. Mutations of the flavonol-specific regulators *AtMYB11*, *AtMYB12,* and *AtMYB111* resulted in an abolition of flavonol accumulation in *Arabidopsis* [23]. Functions of MYB regulators are not only conserved in the same species, but also play a similar role in heterologous plants. For example, overexpression of *AtMYB12* and *AtMYB11* in transgenic tobacco or tomato modulated the flavonoid pathway genes and up-regulated flavonol content [7,39]. Overexpression of *GtMYBP3* and *GtMYBP4* from gentian in tobacco promoted flavonol synthesis, resulting in a pink-white petal phenotype [37]. In the present study, the expression trends of *PqMYBF1*, *PqCHS*, *PqCHI*, *PqF3H*, and *PqFLS* were consistent with the accumulation of flavonol in *P.qiui* leaves (Figure 3). Up-regulation of flavonol synthesis genes (*NtCHS, NtCHI, NtF3H*, and *NtFLS*) was detected in *PqMYBF1*-overexpressing tobacco and flavonol accumulation was increased in flowers and leaves of transgenic tobacco (Figure 4C and Figure 5). Based on this, we reasoned that a similar function of PqMYBF1 should exist in tree peony.

Flavonol and anthocyanin biosynthesis are two important branches of flavonoid pathway. Flavonol and anthocyanin share the early biosynthetic pathway of flavonoids. CHS uses coumaroyl-CoA and malonyl-CoA as substrates to form naringenin chalcone, which was converted naringenin by CHI. F3H converts naringenin to dihydroflavonols. Then, dihydroflavonols are converted to flavonols by FLS. At the same time, dihydroflavonols can also formed to leucoanthocyanins by DFR, and then leucoanthocyanins are converted to anthocyanins by ANS. Therefore, DFR and FLS target the same substrate (dihydroflavonol). The competition relationship between them directly affects the accumulation of anthocyanins and flavonols [5,40]. In this study, overexpression of *PqMYBF1* in tobacco significantly up-regulated the expression levels of *NtCHS, NtF3H,* and *NtFLS* and slightly down-regulated the expression levels of *NtDFR* and *NtANS*, which affected the direction of metabolic flux from anthocyanin to flavonol pathway and caused the accumulation of flavonols and the reduction of anthocyanins, resulting in a pink-white petal phenotype in transgenic tobacco (Figure 4). Similar results were also reported in AtMYB12, AtMYB11, and FeMYBF1 [41,42].

AtMYB11, AtMYB12, and AtMYB11 identified in Arabidopsis exerted their regulatory function on the biosynthesis of flavonol in the root and cotyledons by transactivation of *CHS*, *CHI*, *F3H,* and *FLS* [7,19,36]. AtMYB21 in Arabidopsis stamen regulated flavonol accumulation by increasing *AtFLS1* promoter activity [43]. In *M. domestica*, MYB22 directly binded to *FLS* promoter and affect flavonol biosynthesis [44,45]. In gerbera, GhMY1a significantly activated the promoter of *NtCHS* and *NtFLS*, causing an increase in flavonol accumulation in tobacco [24]. A study of *F.esculentum* reported trans-activation function of the flavonol modulator FeMYBF1 on *FeCHS*, *FeFLS* and *FeFLS1* [41]. In *M. truncatula*, *MtMYB134* regulate flavonol biosynthesis by interacting with the promoters of *MtCHS* and *MtFLS* [23]. In our study, PqMYBF1 positively regulated flavonol biosynthesis by activating the activity of *PqCHS*, *PqF3H*, and *PqFLS* promoters (Figure 7). All these results revealed that the regulation mechanism of flavonol biosynthesis is species-specific. In addition, some MYB TFs are flavonol-specific regulators, and they only control flavonol accumulation [5,7,9,19,21,23,25], while other MYB TFs not only regulate flavonol biosynthesis, but also anthocyanin or proanthocyanidin biosynthesis [24,40,46,47]. Differences of promoter sequence in structural genes in different species may be responsible for these different results [7,38].

## 4. Materials and Methods

### 4.1. Plant Materials 

The samples of *P. qiui* were collected from the tree peony germplasm garden of Northwest A&F University at 9:00–11:00 am during the March and April March 2021 (10–18 °C in the day and 5–8 °C in the night), including the leaves, at five different leaf color stages (S1, S2, S3, S4, and S5) (Figure 3A). The collected leaves were immediately frozen with liquid nitrogen and stored at −80 °C for subsequent use. *Nicotiana tabacum* and *Nicotiana benthamiana* were cultivated in the climate chamber at an ambient temperature of 22–25 °C, a humidity of 60–70%, and a cycle of 16 h of light/8 h of darkness.

### 4.2. Cloning and Bioinformatics Analysis of PqMYBF1

Previous studies have shown that tree peony and *V. vinifera* are closer in terms of genetic relationship [30]. To identify flavonol-related transcription factor in *P. qiui*, we downloaded a protein sequence named VvMYBF1 (ACT88298) of grape (*V. vinifera*) which positively regulates flavonol biosynthesis from the National Center for Biotechnology Information (NCBI) GenBank database [8]. Then, VvMYBF1 was used to query against the transcriptome data of *P. qiui* leaves using the BLAST program. One query had the best hit, and was designated as PqMYBF1. The full length of *PqMYBF1* was cloned from the cDNA of *P.qiui* leaves using specific primers (primer sequences were designed using Oligo7.0 software and listed in Appendix A). The complete open reading frame (ORF) of *PqMYBF1* was integrated into the pCAMBIA1300 vector, and the isolation and sequencing of the recombinant plasmid was completed at Tsingke Biotech Beijing China. MYB proteins known to be associated with flavonoid synthesis were retrieved from NCBI and constructed the phylogenetic tree of PqMYBF1 and other known flavonoid synthesis-related MYB proteins using the neighbor-joining method of MEGA 7.0 software. Motif Scan was used to predict the conserved domain. Multiple sequence alignment was performed using DNAMAN 8.0.

### 4.3. RNA Isolation and qRT-PCR

Total RNA were extracted from *P.qiui* using the RNA prep Pure Plant kit (Tiangen Biotech Co. Ltd., Beijing, China). After testing the quality and concentration of total RNA, genomic DNA were removed using a PrimeScript^®^ RT reagent Kit with gDNA Eraser (DRR047A, Takara, Japan). Then, cDNA was synthesized from 1 μg RNA sample. After diluting cDNA to 200 ng/µg with RNA-free H_2_O, qRT-PCR experiments were performed using SYBR Premix Ex Taq II (DRR041A, Takara, Japan) on StepOnePlus Real-Time PCR system (Applied Biosystems, Foster City, CA, USA). To normalize the expression data, *Ubiquitin* was used as a reference gene. The relative expression levels of genes were calculated using 2^−△△Ct^ comparative threshold cycle (Ct) method. The gene-specific primers used for qRT-PCR analysis were listed in Appendix A. To ensure the accuracy of the data, three biological replicates were performed for each gene.

### 4.4. Tobacco Transformation 

An 1140bp fragment containing PqMYBF1 (open reading frame) ORF is inserted into the overexpression vector pcAMBIA1300 using Xbal and Ncol restriction sites to form a recombinant plasmid pCAMBIA1300-PqMYBF1. The fragment containing a 35S-CaMV promoter, the PqMYBF1 ORF, and a hygromycin resistance gene in the recombinant plasmid were integrated into the tobacco genome using an *Agrobacterium*-mediated leaf disc transformation method. Transgenic tobacco plants were screened in MS medium using 20 mg/L hygromycin, and the positive transformants were further identified by PCR method. Transgenic tobaccos and control lines were grown in the same environment and photographed under the same light conditions.

### 4.5. Subcellular Localization 

The coding sequence of *PqMYBF1* without the termination codon was constructed to the 5′ end of the GFP gene using a homologous recombination kit (Novoprotein, Shanghai, China) in the pCAMBIA2300-GFP (Kana Resistance) vector (Primers were listed in Appendix A). The onion epidermis was disinfected with 70% ethanol, then the inner epidermis was torn and spread flat on MS medium in a sterile environment and cultured for one week. The onion epidermis samples were immersed in the *Agrobacterium* carrying the 35S: GFP-PqMYBF1 vector and the mCherry protein directed to the nucleus localization for at least 12 h dark at 26 °C. The GFP signal and mCherry fluorescent signal were observed using a laser scanning confocal microscope.

### 4.6. Transcriptional Activity Test 

The target vector pGBKT7-PqMYBF1 was constructed using a seamless cloning and assembly kit (Novoprotein, Shanghai, China), and the primer sequences are shown in Appendix A. The PGBKT7-PqMYBF1 vector and a negative control of pGBKT7 vector only containing the BD domain of GAL protein were introduced into yeast Y2H competent cells, respectively. Monoclonal yeast plaques of pGBKT7-PqMYBF1 and pGBKT7 plasmids were picked and dissolved in 200 µL 0.09% Nacl solution and adjusted to be OD_600_ = 1. The absorbed dilutions were incubated on SD/-Trp, SD/-Trp + 40 µg/mL X-α-Gal (Coolaber, Beijing, China), SD/-Trp + 40µg/mL X-α-Gal + 200 ng/mL AbA (Clontech, Mountain View, CA, USA) medium at 29 °C for about 72 h to observe the formation of plaque and coloration.

### 4.7. GUS Reporter Assay 

The promoters of *PqCHS*, *PqF3H*, and *PqFLS* were inserted into pBI121-GUS to activate the GUS reporter gene. Primers are listed in Appendix A. The recombinant vectors were introduced into *Agrobacterium tumefaciens* strain GV3101. Agrobacterium cultures carrying the empty vector pBI121 served as a positive control. Bacterial liquids carried positive controls. Agrobacterium cultures carrying the recombinant vector were injected into tobacco leaves, respectively. GUS staining analysis was as previously described [48].

### 4.8. Dual-Luciferase Reporter Assay

The promoter sequences of the *PqCHS*, *PqF3H*, and *PqFLS* genes were cloned and inserted into the pGreenII-0800-LUC vector to construct the reporter vector, and the effector vector was constructed by inserting the *PqMYBF1* ORF into PGreen-62-SK vector. Primers used above are listed in Appendix A. The effector vector and reporter vector were, respectively, combined and introduced into *A. tumefaciens* strain, then the bacterial liquids were injected into tobacco leaves (*N. benthamiana*) for transient expression analysis. The control group consisted of the pGREENII-62-SK vector combined with a reporter vector containing promoter sequence. The fluorescence expression status of firefly luciferase and renilla luciferase in tobacco leaves was detected under a promega luminometer (Promega, Madison, WI, USA), and the LUC/REN fluorescence ratio was calculated. Three biological replicates were performed for each sample group.

### 4.9. Measurement of Anthocyanin and Flavonol Content

Fully open flowers were collected for anthocyanin extraction from wild type and transgenic tobacco lines. The petals were cooled immediately using liquid nitrogen to prevent browning. Then, 25 mg samples were ground into powder in liquid nitrogen, to which were then added 250 µL of methanol and 1% HCL (*v*/*v*), and were then extracted at 4 °C for 24 h. Mixtures were centrifuged at 10,000 rpm for 10 min at 4 °C, and the supernatants were collected. Absorbance of the supernatant was measured at 530 and 657 nm by a spectrophotometer. The relative anthocyanin levels were calculated by a formula of (A530 nm–0.33 A657 nm)/mg. Flavonol was extracted in the same way using a methanol solution, and the supernatants were filtered through 0.22-μm filter membrane. The relative flavonol level was calculated by the formula (A350–A650)/mg. HPLC analysis were carried out to measure flavonol content in tobacco according to the method described previously [49]. Three samples were collected from each plant, and each sample was measured in triplicate.

## 5. Conclusions

PqMYBF1 is an R2R3-MYB transcription factor that has the SG7 and SG7-2 motifs, which are characteristics of flavonol biosynthesis activators. PqMYBF1 could promote flavonol biosynthesis and accumulation by activating the promoter activity of *PqCHS*, *PqF3H*, and *PqFLS*, and promoting their expression. Our results shows that PqMYBF1 is a functional flavonol-specific TF. These findings will deepen our understanding about tree peony leaf color transformation in spring.

## Figures and Tables

**Figure 1 plants-12-01427-f001:**
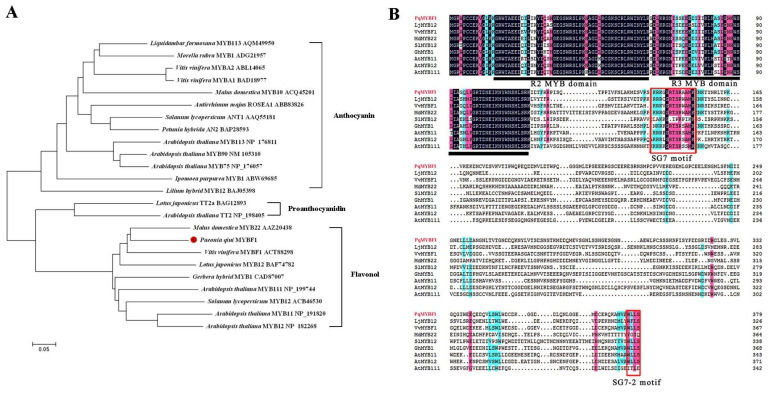
Phylogenetic analysis and sequence alignment of PqMYBF1 with other MYB proteins (**A**) Phylogenetic analysis of PqMYBF1 protein and MYB proteins from other species. Presumed functions of these MYB proteins are listed on the right side of the phylogenetic tree. PqMYBF1 is highlighted with a red dot. The neighbor-joining method with MEGA software was used to construct the phylogenic tree. Bootstrap values as a percentage of 1000 replicates are indicated at corresponding branch nodes. Scale bar represents the number of amino acid substitutions per site. (**B**) Sequence alignment of PqMYBF1 with MYB involved in flavonol synthesis. The black line indicates the positions of the R2 and R3 MYB domains. The red box shows the SG7 and SG7-2 motif.

**Figure 2 plants-12-01427-f002:**
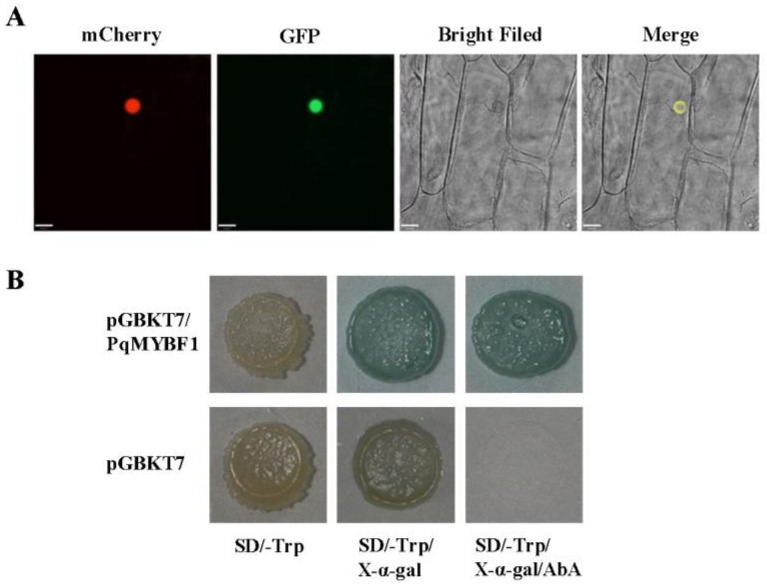
Subcellular localization and transcriptional activity of PqMYBF1. (**A**) Subcellular location of GFP fusion of PqMYBF1. The mCherry protein indicates nucleus localization. Bars = 33 µm. (**B**) Transcriptional activity of PqMYBF1 in yeast. Y2H transformed with pGBKT7-*PqMYBF1* or pGBKT7 vector was grown on SD/-Trp, SD/-Trp with 40 µg/mL X-α-gal and SD/-Trp with 40 µg/mL X-α-gal adding 200 ng/mL aureobasidin A (AbA) for three days.

**Figure 3 plants-12-01427-f003:**
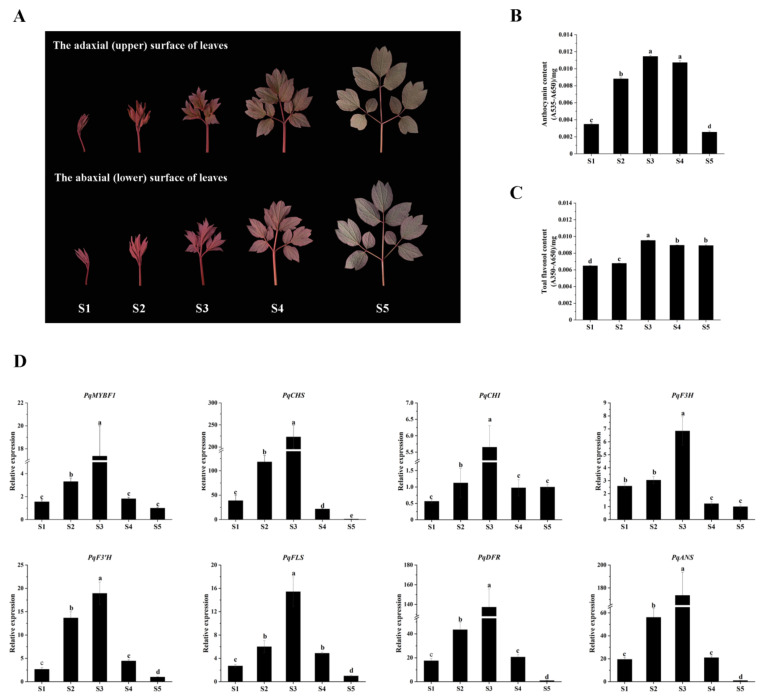
Leaf phenotypes in different states of *Paeonia qiui* and expression level of flavonoid structural genes and *PqMYBF1* at different stages in *P. qiui*. (**A**) The adaxial (upper) and abaxial (lower) surface of *P. qiui* leaves at different stages. The leaf development was divided into five stages mainly based on leaf status and pigmentation: S1: the leaves curled up; S2: the leaves extended slightly; S3: the leaves basically unfolded; S4: the leaves unfolded completely; S5: the leaves turned green. (**B**) Anthocyanin content in *P. qiui*. (**C**) Total flavonol content in *P. qiui*. (**D**) The expression level of *PqMYBF1*, *PqCHS*, *PqCHI*, *PqF3H*, *PqF3′H*, *PqFLS, PqDFR,* and *PqANS*. a, b, c, and d indicate significant differences at the *p* ≤ 0.05 level in the Duncan test.

**Figure 4 plants-12-01427-f004:**
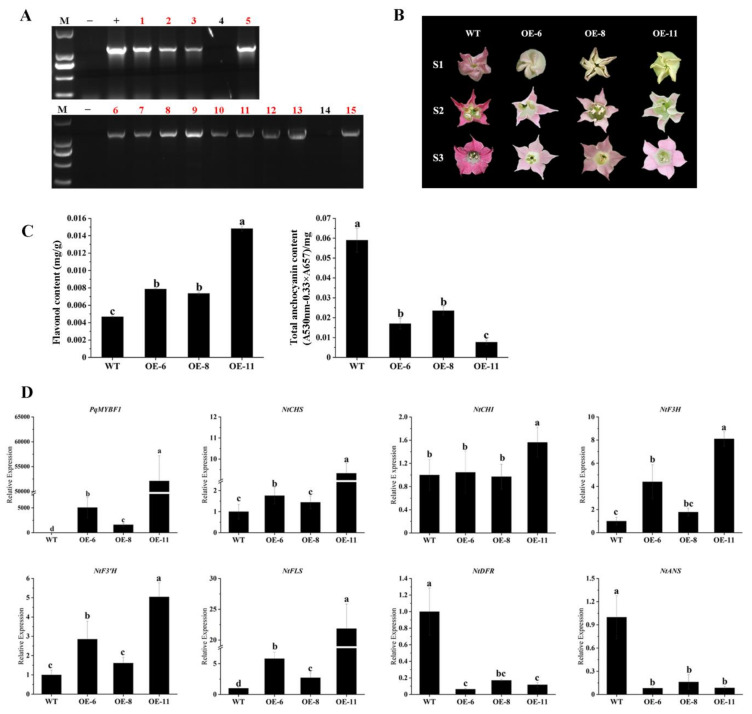
The phenotype and the effect of PqMYBF1 overexpression in transgenic tobacco. (**A**) PCR analysis of PqMYBF1 transformed tobaccos. M: Trans2K Plus DNA Marker; −: Negative control; +: Positive control; 1–15: PCR identification of tobacco; the red numbers indicate the transgenic tobaccos. (**B**) The petal phenotypes of transgenic and wild tobaccos at three flowering stages. S1: closed buds, S2: initially open flowers, S3: open flowers. (**C**) The anthocyanin and flavonol content in petals. (**D**) The expression profiles of vital anthocyanin and flavonol biosynthesis pathway genes in *PqMYBF1* overexpression transgenic tobacco petals. a, b, c, and d indicate significant differences at the *p* ≤ 0.05 level in the Duncan test.

**Figure 5 plants-12-01427-f005:**
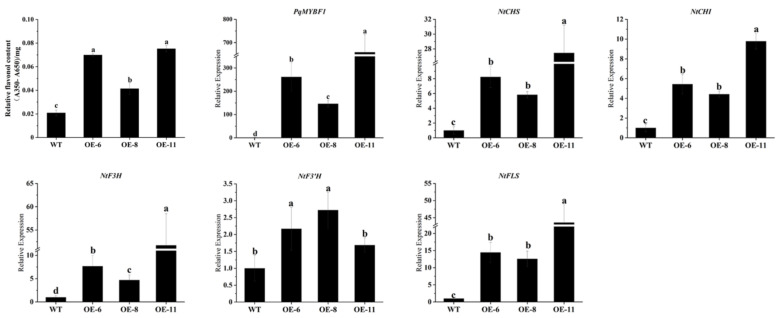
The expression profiles of vital flavonol biosynthesis pathway genes in *PqMYBF1* overexpression transgenic tobacco leaves. a, b, c, and d indicate significant differences at the *p* ≤ 0.05 level in the Duncan test.

**Figure 6 plants-12-01427-f006:**
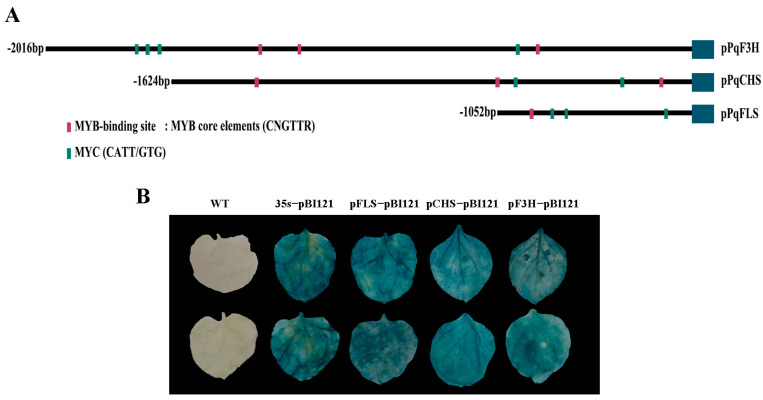
Promoter analysis of flavonol synthesis-related gene. (**A**) The distribution of MYB binding elements in the promoter sequences of *PqCHS, PqF3H,* and *PqFLS*. (**B**) GUS activity analysis of promoters from *PqCHS*, *PqF3H*, and *PqFLS*.

**Figure 7 plants-12-01427-f007:**
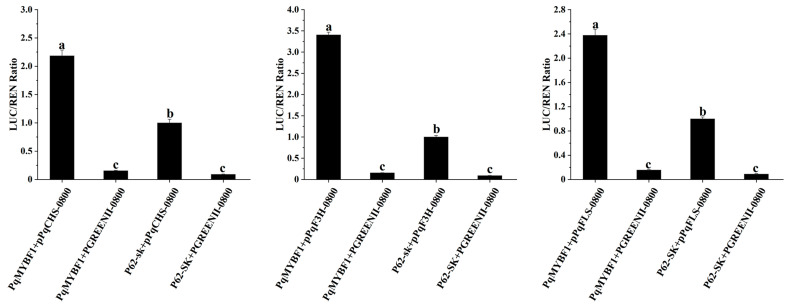
PqMYBF1 interacts with the flavonol-related synthesis gene in vivo. PqMYBF1 induced the expression of *PqCHS*, *PqF3H*, and *PqFLS*. The activities of promoters were indicated by the ratio of LUC/REN. Three independent experiments were performed for each sample. The data are shown as the means ±SDs. a, b, and c indicate significant differences at the *p* ≤ 0.05 level in the Duncan test.

## Data Availability

The datasets generated during and/or analyzed during the current study are available from the corresponding author on reasonable request.

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
