# Peer review of "The Paeonia qiui R2R3-MYB Transcription Factor PqMYBF1 Positively Regulates Flavonol Accumulation"

_plants, 2023, doi:10.3390/plants12071427_

Round 1
Reviewer 1 Report
In this manuscript (plants-2251422) entitled "The Paeonia qiui R2R3-MYB Transcription Factor PqMYBF1 Positively Regulates Flavonol Accumulation" submitted to Plants, authors characterized PqMYBF1, one flavonol biosynthesis related MYB gene in tree peony. Subcellular localization and transactivation assay showed that PqMYBF1 localized to the nucleus and acted as a transcriptional activator. The ectopic expression of PqMYBF1 in transgenic tabacco caused an observable increase in flavonol level and the anthocyanin accumulation was decreased significantly, resulting in pale-pink flowers. Dual-luciferase reporter assays showed that PqMYBF1 could activate the promoters of PqCHS, PqF3H, and PqFLS. These results suggested that PqMYBF1 could promote flavonol biosynthesis by activating PqCHS, PqF3H, and PqFLS expression, which lead metabolic flux from anthocyanin to flavonol pathway, resulting in more flavonol accumulation. The data are convincing and the writing is clear and straightforward. However, some issues needs to be addressed for improving the quality of this manuscript.
1, For Figure 1, I get nothing new from this figure, please remove or list this figure in supplemental figure in the revision.
2, For Figure 2, methods for phylogenetic analysis should be described in the revised legend.
3, For Figure 3a, what is the mcherry? The X-gal signal in Figure 3b is very week, please quantify in the revision.
4, For Figure 4a and Figure 5b, please label stages for plant materials.
5, Please double-check the reference list. For instance, both journal abbreviations and full names appeared.
Author Response
请参阅附件。

Reviewer 2 Report
With their large, showy and colorful flowers, tree peonies have important ornamental value. However, the color of tree peony leaves also changes. In this paper, the authors analyzed PqMYBF1 gene and transformed tabacco. However, the manuscript is very poor and there are some major problems as follows:
1. Figure 1 is not the results of this study. It is not suitable here and suggest to delete it.
2. In most of the figures, many pictures and charts are not properly laid out, making it difficult for readers to understand them clearly. Fig. 2 should show the full name of species in the legend.
3. The logical mess was found in Results 2.1. The authors conducted the multiple sequence alignment analysis first, while in the Figure 2, the authors put the figure of phylogenetic analysis first, please unify the order of text and figures.
4. In Figure 3a, the pCAMBIA1301-GFP empty vector was missing as a control. Should complement it. In addition, mCherry should be explained.
5. In Figure 4 and 6, the font size in bar charts is too small. In fig.6, the author should provide expression level of PqMYBF1 in different lines such as OE6,8,11.
6. In the previous figures of the manuscript, the authors used “a, b, c and d” to indicate the significant differences, why the authors changed to use “*” to indicate significant differences?
7. Some bar charts in this manuscript have vertical coordinates starting from “0” and some starting from “0.0”, please standardize their format.
8. The author conducted the promoters of PqCHS, PqF3H, and PqFLS genes have biological activity. What does author want to demonstrate?
Reviewer 3 Report
The manuscript of paper by Zhang Y. et al. “The Paeonia qiui R2R3-MYB transcription factor PqMYBF1 positively regulates flavonol accumulation” provides new experimental data on the regulation of flavonol and anthocyanin biosynthesis in flowering plants. Authors clearly showed that PqMYBF1 could be a potential regulator of flavonol biosynthesis not only in host plant, P. qiui, but also in tobacco plants. Promoter structure of PqMYBF1 as well as encoded protein was analyzed. And the regulation of flavonol and anthocyanin biosynthesis genes was characterized. The manuscript is written in rather good English, but it contains some inaccuracies throughout the text. The major comments are:
1) Line 11: FlavonolS ARE one subgroup of flavonoids
2) Lines 13, 17, 156, 255, 416, 439, 442, 445,463, 465-466, 469, 474, 476, 493, 495, 498, 500, 505, 512, 518, 536, 545: genes and plant Latin names have to be in Italics, proteins have to be in Regular. Full Latin names in the format Genus species should be given at the first mention in the text, as well as in the title, abstract and methodology, otherwise it has to be G. species. Short name should not appear before the full name (lines 365 and 374).
3) Line 21: “… which leadS metabolic …”.
4) There are several missing spacebars (lines 39, 66, 126, 149, 150) as well as excessive dot (line 41) and dash (line 53).
5) Figure 2a, not 2b, should be discussed first in the text (lines 91-102). Both the text and the figure should be rearranged in accordance with this.
6) Figure 5b: It seems that the flowers from the top row to the bottom raw have undergone development, so it is necessary to specify the age.
Taking all mentioned above into account, the manuscript would merit publishing in Plans, but it requires minor text revision. After the correction it can be recommended for publishing.
Round 2
Reviewer 1 Report
Authors have addressed my concerns in the revision.
Reviewer 2 Report
I think the authors basically answer the questions and revise the manuscript.